# Altered Cell Surface N-Glycosylation of Resting and Activated T Cells in Systemic Lupus Erythematosus

**DOI:** 10.3390/ijms20184455

**Published:** 2019-09-10

**Authors:** Enikő Szabó, Ákos Hornung, Éva Monostori, Márta Bocskai, Ágnes Czibula, László Kovács

**Affiliations:** 1Institute of Genetics, Biological Research Centre of the Hungarian Academy of Sciences 6726 Szeged, Hungary; 2Department of Rheumatology and Immunology, Faculty of Medicine, University of Szeged, 6725 Szeged, Hungary

**Keywords:** systemic lupus erythematosus, T cells, glycosylation, sialylation, lectin binding, glycosylation enzymes, galectin 1

## Abstract

Altered cell surface glycosylation in congenital and acquired diseases has been shown to affect cell differentiation and cellular responses to external signals. Hence, it may have an important role in immune regulation; however, T cell surface glycosylation has not been studied in systemic lupus erythematosus (SLE), a prototype of autoimmune diseases. Analysis of the glycosylation of T cells from patients suffering from SLE was performed by lectin-binding assay, flow cytometry, and quantitative real-time PCR. The results showed that resting SLE T cells presented an activated-like phenotype in terms of their glycosylation pattern. Additionally, activated SLE T cells bound significantly less galectin-1 (Gal-1), an important immunoregulatory lectin, while other lectins bound similarly to the controls. Differential lectin binding, specifically Gal-1, to SLE T cells was explained by the increased gene expression ratio of sialyltransferases and neuraminidase 1 (*NEU1*), particularly by elevated ST6 beta-galactosamide alpha-2,6-sialyltranferase 1 (*ST6GAL1*)/*NEU1* and ST3 beta-galactoside alpha-2,3-sialyltransferase 6 (S*T3GAL6)*/*NEU1* ratios. These findings indicated an increased terminal sialylation. Indeed, neuraminidase treatment of cells resulted in the increase of Gal-1 binding. Altered T cell surface glycosylation may predispose the cells to resistance to the immunoregulatory effects of Gal-1, and may thus contribute to the pathomechanism of SLE.

## 1. Introduction

Numerous congenital and acquired diseases show altered cell surface glycosylation, including several types of cancer and autoimmune syndromes [1,2]. Altered oligosaccharide structures have been identified in tumors and have proven to be diagnostic markers of malignant phenotypes [1,3,4]. Protein glycosylation has become an integral part of research in autoimmunity, as defective glycan structures have been described on serum immunoglobulins [5] and the different glycans at certain residues on IgG subclasses affected the effector function of autoantibodies [6,7]. Surface glycosylation of immune cells has also been studied, and glycoconjugates have been proven to play a role in many fields of cellular physiology, such as migration and signal transduction [8]. T cell functions can also be modulated by interaction between cell surface glycoproteins and endogenous lectins, including galectins [9].

Glycoconjugates are created in the endoplasmic reticulum, the Golgi apparatus, and on the cell surface by enzymes, including mannosidases, glycosyltransferases, sialyltransferases, and neuraminidases (NEU) [10]. The concerted action of these enzymes produces the specific sugar ‘code’ presented on the cell surface, which then regulates further signaling and adhesion properties of a particular cell type. The expression of enzymes participating in glycosylation can determine the sensitivity of the cells to numerous extracellular signals.

Galectin 1 (Gal-1) is a member of the β-galactoside binding mammalian lectin family with specific affinity to terminal N-acetyllactosamine motifs on multi-antennary cell surface glycans [11]. One of the major effects of Gal-1 in immunoregulation is the induction of apoptosis of the activated T cell subpopulations Th1 and Th17, whereas Th2 and Treg cell functions are promoted by Gal-1 [12,13]. This selectivity is caused by the differences in surface glycosylation of various T cell subtypes [9]. Gal-1-triggered cell death has been extensively studied in vitro, and its mechanism has been described [14,15,16,17,18]. Lactosamine sequences, required for Gal-1 binding, are synthesized by specific glycosyltransferases, such as beta-N-acetylglucosaminyltransferases and beta-galactosyltransferases. The expression of such enzymes controls T cell susceptibility to Gal-1-driven apoptosis [19].

We have recently demonstrated that activated T cells from patients with active systemic lupus erythematosus (SLE) are resistant to the apoptotic effect of Gal-1 [20], and we suggested that this finding is relevant to the immunoregulatory dysfunction observed in SLE. As a potential cause of this resistance may be an impaired binding of Gal-1 to the T cell surface, we set out to examine cell surface glycosylation and the expression of glycosylation enzymes in SLE T cells in comparison with healthy control T cells. The glycosylation pattern of resting SLE T cells resembled the activated phenotype of T cells. Activated SLE T cells bound significantly less galectin 1 (Gal-1) than the controls, while other lectins bound similarly. To understand the distinct lectin binding, specifically Gal-1, to SLE T cells, we found that the terminal sialylation increased in the autoimmune cells, and accordingly, neuraminidase treatment resulted in a remarkable increase in Gal-1 binding.

## 2. Results

The N-glycome diversity of T cell surface glycans was analyzed by the binding of lectins derived from plants (concanavalin-A (ConA), *Lens culinaris* agglutinin (LCA), wheat germ agglutinin (WGA), *Phaseolus vulgaris* leukoagglutinin (PHA-L), and *Sambucus nigra* agglutinin (SNA)) or of a human lectin, Gal-1, with known sugar binding specificity (Appendix A and Table 1). Lectin binding to resting and phytohaemagglutinin (PHA)-activated T cells obtained from SLE patients and healthy controls was measured. Analysis of resting T cells from SLE patients and control individuals revealed that resting SLE T cells bound significantly more ConA, LCA, and WGA than healthy T cells (Figure 1A). ConA couples with mannoses present in early high-mannose glycans and mannoses in complex sugars [21,22], while LCA has high affinity to fucosylated core mannoses present in bi-antennary complex N-glycans and does not bind to tri- and tetra-antennary N-glycans [23]. WGA binds to N-acetyl glucosamines present in hybrid-type sugar chains (early and complex sugars) or to sialic acid, which can be terminally attached to complex multi-antennary glycans, and its affinity to the sialylated version of tri- or tetra-antennary glycan-containing glycoproteins was shown to be higher than to the desialylated form [24,25].

Comparing healthy and autoimmune-activated T cells, we found that activated SLE T cells bound lectins in levels similar to control cells with the exception of Gal-1. SLE cells bound significantly less Gal-1 than control cells, indicating that terminal N-acetyllactosamine side chains, the Gal-1 ligands, were less accessible on these cells (Figure 1B). The changes in the pattern of lectin bindings did not occur preferentially on either CD4+ or CD4- (CD8+) cells, as these were similar in the control as well as in SLE activated T cells (Appendix A).

Glycosylation of proteins is regulated by multiple factors in the Golgi apparatus, such as sub-Golgi localization of glycosylation enzymes, transporters, pH, endoplasmic reticulum stress, or substrate availability (reviewed in [29]). However, a major element is the expression and function of glycosylation enzymes [30,31]. Therefore, expression levels of the genes involved in N-linked glycosylation (Appendix A and Table 2) were examined by qPCR analysis of activated T cells. Gene expression of alpha mannosidases (*MAN1A1*, *MAN1A2*, *MAN2A1* and *MAN2A2*) in activated SLE T cells did not differ from the controls (Figure 2A). Analysis of beta-N-acetylglucosaminyltransferases (*MGAT1–5*) presented a slight but significant difference in the cases of *MGAT4A* and *MGAT4B* (Figure 2B).

Poly-N-acetyllactosamine chains on N glycans can be capped with the attachment of α-2,6 sialic acid by ST6 beta-galactosamidealpha-2,6-sialyltranferase 1 (*ST6GAL1*) and α-2,3 sialic acid by *ST3GAL3*, *ST3GAL4*, and *ST3GAL6* [32],and cleaved by neuraminidases. In the control and patient groups, *ST6GAL1*, *ST3GAL3*, *ST3GAL4*, and neuraminidase 1 (*NEU1*) gene expression levels were similar, whereas the mRNA level of *ST3GAL6* was significantly elevated in SLE T cells (Figure 3A). During T cell activation, the gene expression of *NEU1* is strongly upregulated, while *NEU3* expression remains constant [33]; hence, only *NEU1* was analyzed.

Concerted action of sialyltransferases and neuraminidases determine the sialylation pattern. The large variations between the Gal-1 binding of the control and SLE T cells (Figure 1B) indicated an alteration in the sialylation of SLE surface glycans. As this is determined by the net effect of enzymes that sialylate (sialyltransferase) and desialylate (neuraminidase) the glycans, gene expression ratios of the opposing acting enzymes were calculated. A significantly higher *ST3GAL6*/*NEU1* and *ST6GAL1*/*NEU1* mRNA ratio was observed in SLE compared to control T cells (Figure 3B), indicating higher sialylation of SLE T cells. Other sialyltransferase/neuraminidase mRNA ratios, such as *ST3GAL3*/*NEU1* and*ST3GAL4*/*NEU1*, remained similar in the control and SLE groups (Figure 3B). These results indicated that reduced Gal-1 binding to SLE T cells may be a result of a more densely sialylated glycan profile. Indeed, cleaving sialic acid from the surface of SLE activated T cells by α2-3,6,8 Neu (specific to α2-3,6,8 linked sialic acid) resulting in the elevation of Gal-1 binding (Figure 4). The increase of Gal-1 binding to SLE cells was similar to that of control T cells (data not shown).

## 3. Discussion

Selected steps of mammalian N linked glycosylation and lectin binding to specific sugar side chains are summarized in Appendix A. It must be noted that binding of the used lectins was more degenerated than what is shown in the simplified Appendix A; however, it may help in a better apprehension of this work.

Remarkable differences were detected in ConA, LCA, and WGA binding between resting SLE and control T cells, since SLE T cells bound significantly higher amounts of these lectins. As LCA and WGA recognize matured sugar side chains and ConA couples both unmatured (early) and matured (complex) glycans, these results indicated that resting SLE T cells present a glycan structure similar to their activated phenotype. Detection of other activation markers, such as heightened CD40L expression [34], CD44 expression [35,36,37], exhibition of constant membrane raft polarization and increased GM1 content [38], measured by others, also suggest similarity to an activated state. On the other hand, this activated phenotype did not manifest in terms of CD25 expression on resting SLE T cells, as CD25 levels were similar on resting and activated T cells, suggesting that the activated phenotype is limited to several, but not all activation markers (Appendix A).

Stimulation of control T cells with PHA-L resulted in an elevation of binding of all used lectins, except the terminal α-2,6 sialic acid binding SNA (data not shown), indicating a generally increased complexity of glycosylation pattern upon activation. These results were in accordance with previous findings, arguing that N-glycan abundance, branching, glycan chain elongation, and hereby complexity enhanced [39,40], and terminal α-2,6 sialic acid residues declined [41,42,43] on freshly activated T cells. However, the increase in glycan complexity after activation was hardly seen in SLE T cells, as an increase in lectin binding upon activation was rather low or was absent in SLE, a phenomenon that is also explained by the activated phenotype of SLE T cells, even without treatment with activating agents. Furthermore, the decreased Gal-1 binding of SLE T cells was observed, not only in resting state, but also persisting after activation. No difference was found between CD4+ and CD4- (CD8) cells in terms of binding of any of the lectins (Appendix A). Some previous findings indicate that the proportion of effector memory (C-C Motif Chemokine Receptor 7{CCR7}-CD27+) and terminally differentiated effector memory (CCR7-CD27-) cells increase in SLE, and may correlate with disease activity or damage [44,45]. It would be interesting to compare the cell surface glycosylation patterns of naive and various subtypes of memory T cells; however, it was outside the scope of our present study. It is also to be noted that SLE memory effector T cells are crippled in response to antigen stimulation, as they respond to stimulation with apoptosis instead of proliferation [45], which might be a consequence of the altered glycosylation.

Variation in the binding of lectins tested was only confined to Gal-1, whose binding is determined by the presence of asialylated terminal N-acetyl lactose residues [46]. To clarify the background of this specific variability of Gal-1 binding, the expression of enzymes involved in creating the glycosylation pattern was determined. Analysis of the mRNA expression of glycosylation enzymes was chosen, since previous data indicated that glycosylation was primarily regulated at the transcriptional level of the appropriate enzymes [31]. Expressions of glycosyltransferases, alpha mannosidases, beta-N-acetylglucosaminetransferases, sialytransferases, and a neuraminidase, *NEU1*, were similar in control and SLE T cells. These findings are in accordance with the results of lectin assays, as all lectins tested bound similarly to the activated control and autoimmune T cells, with the exception ofGal-1. The difference in Gal-1 binding between control and patient activated T cells might therefore result from the distinct sialylation of the Gal-1 binding glycoconjugates. This presumption seemed to be supported by the finding that ratios of the expression of sialyltransferases (*ST3GAL6* and *ST6GAL1*) and neuraminidase shifted towards the sialytransferases, indicating a more intensified sialylation of SLE T cell glycans, including Gal-1 binding structures. Sialylation plays an important role in masking terminal carbohydrate chains, hence regulating lectin binding and signal transduction processes. Sialyltransferases attach, while neuraminidases remove sialic acid residues of terminal carbohydrate groups. Consequently, the accessibility of lectin binding sites is specifically regulated by the concerted action of sialyltransferases and neuraminidases. This point of view was supported with the finding, that ablation of sialic acid from surface glycoconjugates of living activated SLE T cells by neuraminidase treatment resulted in an increase in Gal-1 binding.

An important issue is how glycosylation affects autoimmune T cell activation, cell–cell interactions and autoantibody production. The available literature data are limited, and further detailed investigation is required. However, several studies suggest that pathological glycosylation results in disturbed T cell receptor (TCR)-major histocompatibility complex (MHC) interactions [38], cell adhesion [40], necrotic cell death- and glycan-specific autoantibody production [47], and deviant antigen presentation [48,49]. How closely the pathomechanism of SLE is associated with the surface glycosylation pattern and its abnormalities remains to be elucidated. Nevertheless, the present findings seem to corroborate our hypothesis regarding the resistance of activated SLE T cells to the apoptotic effects of Gal-1. As we described earlier [20], activated SLE T cells showed a reduced response to Gal-1 due to their defective expression of intracellular Gal-1. The present work suggests that altered glycosylation and, hence, the decreased binding of extracellular Gal-1 to SLE T cells can be another cause of the resistance to Gal-1-mediated immunomodulation, serving a putative novel pathogenic mechanism in SLE. It has to also be clarified whether removal of sialic acid from cell surface glycoconjugates results in the restoration of Gal-1-induced apoptotic sensitivity. Nevertheless, it has become clear from this work that analyzing the glycosylation process, especially the expression ratio of sialyltransferases and the neuraminidases and the binding of Gal-1 to the cell surface, may emerge as a novel approach to connecting disease phenotypes with functional pathways within T cells (differential-diagnosis of diseases or patient subset analysis within particular multisystem autoimmune diseases).

Since SLE is an autoimmune disease with multiple alterations on genetic, protein, signaling, and glycosylation levels, it is difficult to determine the primary cause of the disorder. It is likely that all the more-or-less relational changes result in the final manifestation of SLE. As glycosylation affects cell migration, adhesion, and signal transduction, its changes must be an important factor contributing to the pathomechanism of the disease. This view is supported by the finding that the binding of Gal-1 to SLE T cells, an anti-inflammatory human lectin, decreases because of the different sialylation of SLE T cells from that of healthy T cells, and thus it must be another reason that SLE T cells are more resistant to Gal-1-induced apoptosis [20].

The glycosylation phenotype resembles an activated state of SLE T cells. Though the fundamental causes behind the development of SLE are unclear, it is known that alpha-mannosidase II knock out mice develop an SLE-like disease [50]. This enzyme removes early mannose from maturing glycoconjugates, therefore, it is crucial to the final formation of healthy complex N-glycan structures observed in mammals. Its deficiency leads to immature, mannose-rich glycan chains being upregulated because of a disruption in their stepwise disassembly before the complex chain can be built in their place [51]. This effect is similar to our findings indicating a higher-than-normal distribution of mannose-rich glycans on SLE T cells before activation. It is known that mannose-rich chains are much more common on many strains of fungi and are easily recognized as non-self structures leading to auto-immune reactions [50].

Altered glycosylation in SLE is not confined to T cells, as the glycan profile of IgG is a primary predictor of the inflammatory capability of the molecule. Asialylated, agalactosylated glycan chains are stronger activators of complement than sialylated chains, and as such, pro-inflammatory responses are upregulated by the IgG molecule if it contains less terminal syalic acid units [5].

Altogether, the above data indicate that alteration in glycosylation in SLE is likely to be a primary phenomenon, and it contributes to the pathomechanism of the disease.

To our knowledge, the current work provides the first evidence for altered T cell surface glycosylation in SLE. Our major findings are that resting SLE T cells show an activated phenotype from the glycosylation point of view and lectin binding of activated SLE T cells is similar to the controls, with exception of the significantly lower Gal-1 binding. Furthermore, this is a consequence of a shift toward terminal sialylation of glycan structures due to an increased *ST6GAL1*/*NEU1* and/or *ST3GAL6*/*NEU1* ratio. Indeed, desialylation of the surface glycans on SLE T cells results in a remarkable increase in Gal-1 binding.

## 4. Materials and Methods

### 4.1. Ethical Statement

The study was designed in accordance with the guidelines of the Declaration of Helsinki and was approved by the Human Investigation Review Board, University of Szeged reference, No. 2833/2011 on 21 February 2011.

### 4.2. Patients

Patients with SLE (*n* = 18) and healthy controls (*n* = 19) were examined, except in one experiment where *n* = 3 (Figure 3). All patients met the 2012 SLICC classification criteria for SLE [52,53] and had active disease, as reflected by relevant disease activity indices. Eligible patients had an SLE Disease Activity-Index-2000 (SLEDAI-2K) ≥ 6 [54], did not have a co-existent inflammatory condition (overlapping autoimmune disease or infection), and did not have diabetes mellitus. Treatment with potent immunosuppressive drugs (mycophenolate mofetil, cyclophosphamide, rituximab) or corticosteroid at a dose >5 mg prednisolone equivalent was also an exclusion criterion. Controls were healthy individuals without any inflammatory disease or diabetes mellitus.

Demographics and the relevant disease activity data are presented in Table 3.

### 4.3. Cells

Peripheral blood mononuclear cells (PBMC) were isolated from SLE patients and healthy donors using Ficoll (GE Healthcare, Amersham, UK) gradient centrifugation. A part of the PBMCs was stimulated with 1 μg/mL phytohaemagglutinin-L (PHA-L, Sigma-Aldrich, St. Louis, MO, USA) and the cells were cultured for 72 h in a humidified incubator with 5% CO2 at 37 °C in Roswell Park Memorial Institute (RPMI)-1640 medium (Gibco, Life Technologies, Paisley, UK) supplemented with 10% fetal bovine serum (FBS) (Gibco, Life Technologies, Paisley, UK), 2 mM L-glutamine (Gibco, Life Technologies, Paisley, UK) and penicillin-streptomycin (Sigma-Aldrich, St. Louis, MO, USA), henceforward referred to as activated T cells. Activated T cell cultures were > 90% pure, as controlled with flow cytometry using anti-human CD3 antibody (BioLegend, San Diego, CA, USA) (data not shown).

### 4.4. Lectin Binding Assay

Resting or activated T cells were washed twice with cold phosphate buffered saline (PBS) and incubated at 4 °C for 30 min with eFluor 660 fixable viability dye (eBioscience, Thermo Fisher Scientific, Waltham, MA, USA), then fixed with 4% paraformaldehyde for 4 min at room temperature. After washing the samples twice in PBS supplemented with 1% FBS and 0.1% sodium-azide (fluorescence-activated cell sorting (FACS) buffer), the cells were incubated at 4 °C for 20 min with fluorescein-labeled lectins or unlabeled Gal-1 (lectins used are listed in Table 1). The fluorescein-labeled plant lectin kit (Vector Laboratories, Burlingame, CA, USA) was used according to the manufacturer’s instructions. Recombinant galectin-1 was produced and characterized in our laboratory, as previously described [15]. Gal-1-binding was detected as follows: after washing with FACS buffer, biotinylated mouse monoclonal antibody to Gal-1 (2C1/6) was added and incubated at 4 °C for 45 min. Samples were washed with cold FACS buffer before adding fluorescein isothiocyanate(FITC)-labeled streptavidin and incubating the cells at 4 °C for 20 min. Finally, the samples were washed twice in FACS buffer. Samples were analyzed with a FACSCalibur system (BD Biosciences, Franklin Lakes, NJ, USA) and data were evaluated using FlowJo V10 software (BD Biosciences, Franklin Lakes, NJ, USA). The lectin binding was evaluated on resting T cells within PBMCs or activated T cells by gating CD3+ cells with PE/Cy5-conjugated anti-human CD3 antibody (BioLegend, San Diego, CA, USA).

### 4.5. Neuraminidase Treatment

Activated T cells (2 × 10^6^/sample) were treated with α2-3,6,8 neuraminidase (α2-3,6,8 Neu, New England BioLabs, Ipswich, MA, USA) according to the manufacturer’s instructions or left untreated.

Briefly, the cells were washed with PBS then incubated with 200 U of α2-3,6,8Neu in a total volume of 40 µL glycobuffer (provided by the manufacturer) for 15 min at 37 °C. After washing the samples with cold PBS, Gal-1 binding, viability staining, fixation, and flow cytometry analysis were done, as described above.

### 4.6. Quantitative Real-Time PCR (qPCR)

The qPCR assays were performed according to the MIQE (Minimum Information for Publication of Quantitative Real-Time PCR Experiments) guidelines [55]. The names of genes are listed in Table 2. Total RNA was extracted from activated T cells (1–3 × 10^6^ cells) using PerfectPure RNA Cultured Cell kit (5 PRIME, Gaithersburg, MD, USA) according to the manufacturer’s instructions with on-column DNase digestion. The amount and quality of RNA were measured using NanoDrop-1000 spectrophotometer (Thermo Fisher Scientific, Waltham, MA, USA). For cDNA synthesis, 2 μg of total RNA/reaction was reverse transcribed using RevertAid H Minus First Strand cDNA Synthesis Kit (Thermo Fisher Scientific, Waltham, MA, USA) in the presence of 1.66 µM of oligo (dT) 18 and random hexamer primers, 0.5 mMdNTP, 10 U RiboLock RNase Inhibitor, and 200 U RevertAid H Minus Reverse Transcriptase for 60 min at 42 °C, then heated for 10 min at 70 °C. Quantitative PCR amplifications were carried out with appropriate negative controls at least in duplicate. The reaction volume was 20 μL and composed of AccuPower 2 × Greenstar qPCR Master Mix (Bioneer, Alameda, CA, USA), 300–300 nM primers, and 40-fold diluted cDNA. For PCR amplifications, RotoGene3000 instrument (Corbett Research, Sydney, Australia) was used. The qPCR program included: initial 95 °C for 15 min, followed by repeated 45 cycles (95 °C for 15 s, 60–62 °C for 20 s, 72 °C for 20 s), and melting temperature analysis increasing the temperature from 55 °C to 98 °C at 0. 5 °C/step with 8 sec stops between each step. Quantitative real-time PCR data were analyzed using the Rotor Gene software (v6.1 build 93). Relative mRNA levels, normalized to *RPL27*, were presented as log_2_ transformation of relative gene expression, namely subtraction of Ct values (Δ*Ct* = Ct*RPL27* − CtGOI). The mRNA expression ratios were calculated using the following formula: Δ*Ct* × gene −Δ*Ct* Y gene. Primer sequences were partly taken from [56,57,58,59], or designed using the Universal Probe Library Assay Design program (Roche Applied Science, Basel, Switzerland). Primer sequences used in the study are shown in Table 4.

### 4.7. Statistical Analysis

Statistical analysis was performed using GraphPad Prism Version 7.01. In all statistical analyses, an unpaired, two-tailed *t*-test was used to compare data obtained in T cells of healthy controls and SLE patients. There was only one exception (Figure 4B), where paired two-tailed Student’s *t*-test was chosen for the comparison of values from T cells of the same patient with or without neuraminidase treatment. The significant differences were indicated as follows: * *p* < 0.05, ** *p* < 0.01, *** *p* < 0.001.

## Figures and Tables

**Figure 1 ijms-20-04455-f001:**
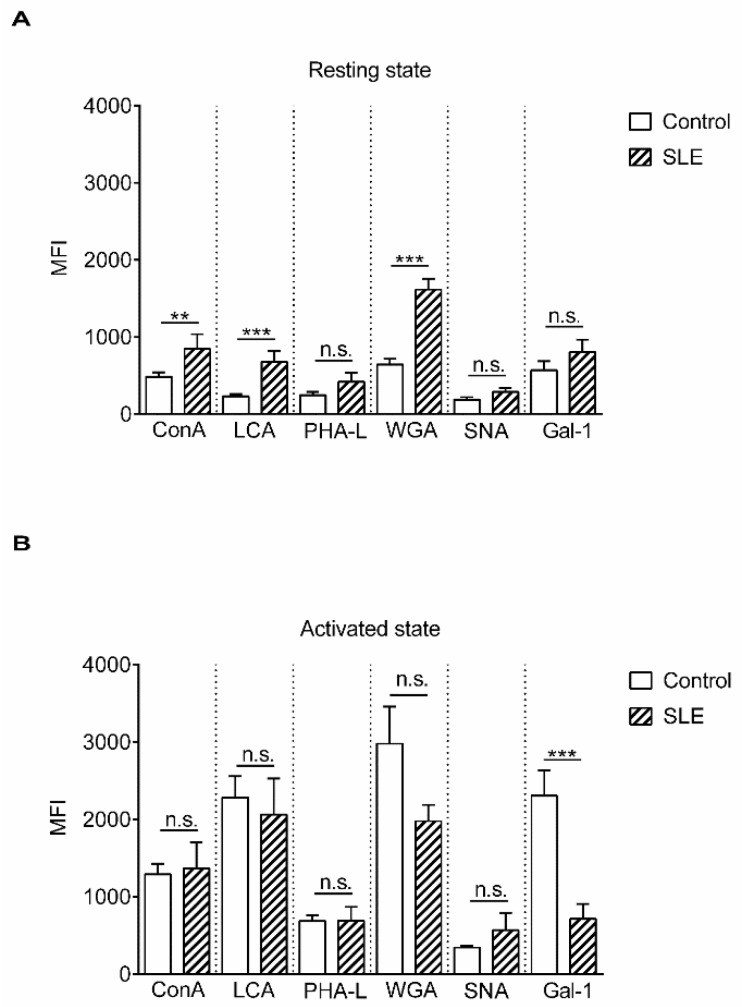
Lectin binding properties of resting and activated T cells from healthy donors and from systemic lupus erythematosus (SLE) patients. Peripheral blood T cells were obtained from healthy controls and SLE patients. The cells were left unstimulated (resting state, **A**) or were activated with 1 µg/mL phytohaemagglutinin L (PHA L) for 72 h (activated state, **B**). Cells were stained with viability dye, fixed then labeled with anti-CD3-PE-Cy5 antibody, followed by fluorescein isothiocyanate (FITC)-conjugated lectin. The samples were evaluated with flow cytometry. Binding of FITC-conjugated lectins is shown as mean (±SEM) of the median fluorescence intensity (MFI) values of flow cytometry histograms of resting (**A**) or activated (**B**) CD3-positive live T cells. Lectin names are listed in Table 1. MFI: mean fluorescence intensity, ConA: concanavalin-A, LCA: *Lens culinaris* agglutinin, WGA: wheat germ agglutinin, PHA-L: *Phaseolus vulgaris* leukoagglutinin, SNA: *Sambucus nigra* agglutinin, Gal-1: galectin 1. Statistical analysis was performed using an unpaired Student *t*-test. ** *p* < *0.01*; *** *p* < 0.001; n. s.: not significant. SLE: *n* = 18, and healthy controls: *n* = 19.

**Figure 2 ijms-20-04455-f002:**
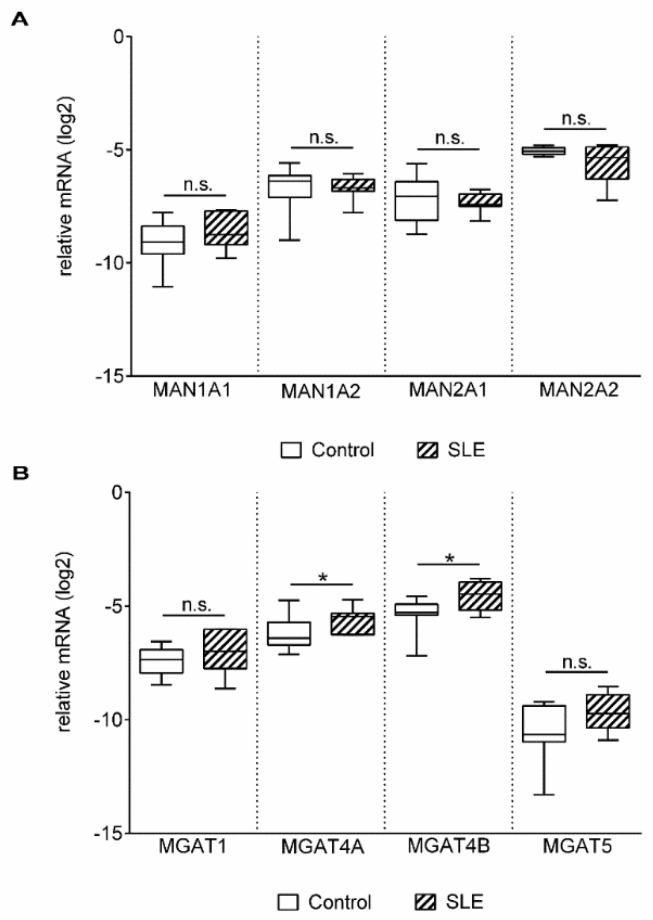
Gene expression of mannosidases (MANs) (**A**) and N-acetyl glucosaminyltransferases (MGATs) (**B**) in activated T cells. Total RNA was extracted from activated T cells and mRNA expression levels were analyzed by qPCR. Results of the relative expression were normalized to the expression levels of the *RPL27* housekeeping gene (log_2_ transformation, Δ*Ct)*. Gene names and primer sequences are listed in Table 2 and Table 4, respectively. Upper and lower quartiles and whiskers of boxes extend to the minimum and maximum values, and the band inside the box is the median. Statistical analysis was performed using an unpaired Student’s *t*-test, where * *p* < 0.05; SLE: *n* = 18, and healthy controls: *n* = 19.

**Figure 3 ijms-20-04455-f003:**
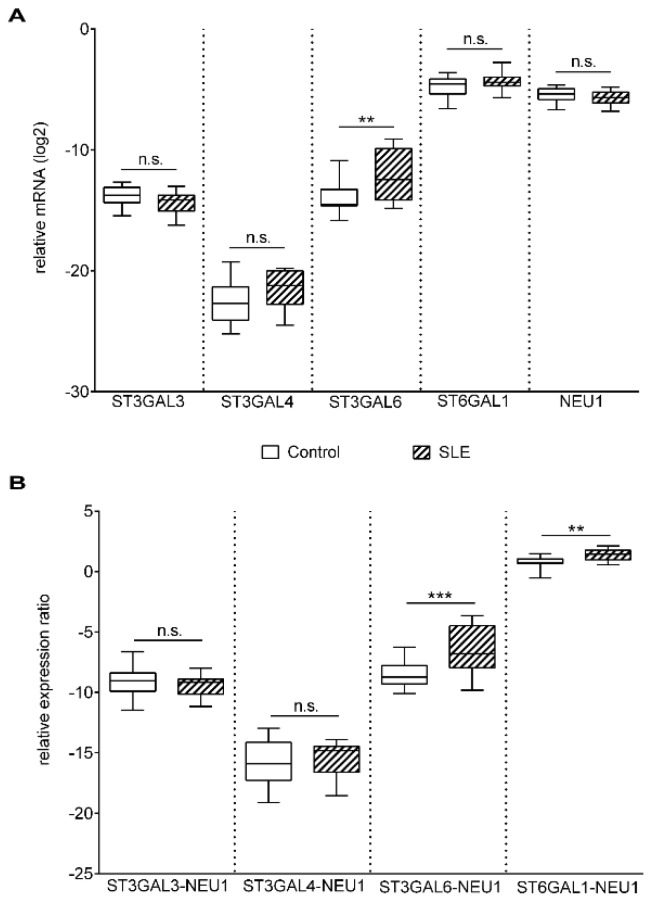
Gene expression of sialyltransferases (ST) and neuraminidase-1 (*NEU1*) in activated T cells. Total RNA was extracted from activated T cells and mRNA expression levels were analyzed by qPCR. (**A**) Results of the relative expression were normalized to the expression levels of *RPL27* housekeeping gene (log_2_ transformation, Δ*Ct*). Gene names and primer sequences are listed in Table 2 and Table 4, respectively. (**B**) ST/*NEU1* mRNA expression ratios. The sialyltransferase-neuraminidase mRNA expression ratios of individual persons were calculated as follows: Δ*Ct* ST/Δ*CtNEU1*. Upper and lower quartiles and whiskers of boxes extend to the minimum and maximum values, and the band inside the box is the median. Statistical analysis was performed using an unpaired Student *t*-test, where ** *p*< 0.01; *** *p*< 0.001. SLE: *n* = 18, and healthy controls: *n* = 19.

**Figure 4 ijms-20-04455-f004:**
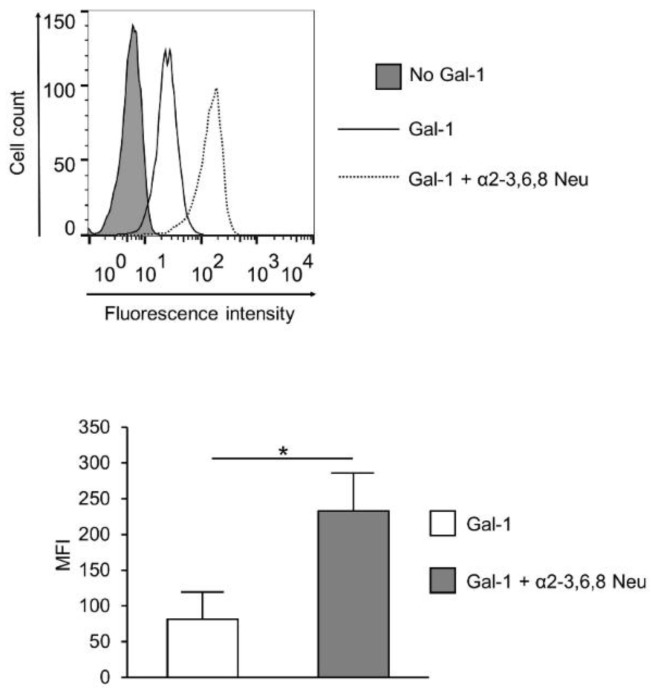
Effect of neuraminidase treatment on Gal-1 binding of SLE T cells. Activated SLE T cells were treated with α2-3,6,8 neuraminidase (Gal-1 + α2-3,6,8 Neu; dotted line) or left untreated (Gal-1; empty, continuous line), and then Gal-1 binding was investigated by cytofluorimetry, as described in the materials and methods section. The grey shadowed histogram shows the negative control: no Gal-1, +streptavidin—FITC. The upper image shows a representative profile of Gal-1 binding, the lower graph shows means (±SEM) of the mean fluorescence intensity (MFI) values of activated SLE T cells. Statistical analysis was performed using a two-tailed paired *t*-test. * *p* < 0.05, *n* = 3.

**Table 1 ijms-20-04455-t001:** Names, abbreviations, and binding specificities of lectins.

Lectins	Abbreviation	Specificity	Reference
Concanavalin A	ConA	mannose, glucose (low affinity)	[21,22]
*Lens culinaris* agglutinin	LCA	core-fucosylated bi-antennary N-glycan	[22,23]
Wheat germ agglutinin	WGA	GlcNAc, sialic acid	[24,25]
*Phaseolus vulgaris* leucoagglutinin	PHA-L	β-1,6-branched tri- and tetra-antennary N-glycans	[26]
*Sambucus nigra* agglutinin	SNA	α-2,6-linked sialic acid	[27]
Galectin-1	Gal-1	LAcNAc	[28]

Abbreviation: GlcNAc: N-acetylgucosamine; LacNAc: N-acetyllactoseamine.

**Table 2 ijms-20-04455-t002:** Symbols and full names of glycosylation enzyme genes.

Enzyme Genes	Gene Symbol	Full Gene Name
Mannosidases	*MAN1A1*	Mannosidase alpha class 1A member 1
	*MAN1A2*	Mannosidase alpha class 1A member 2
	*MAN2A1*	Mannosidase alpha class 2A member 1
	*MAN2A2*	Mannosidase alpha class 2A member 2
N-Acetylglucosaminyltransferase	*MGAT1*	Mannosyl (alpha-1,3-)-glycoprotein beta-1,2-N-acetylglucosaminyl-transferase
	*MGAT4A*	Mannosyl (alpha-1,3-)-glycoprotein beta-1,4-N-acetylglucosaminyl-transferase isozyme A
	*MGAT4B*	Mannosyl (alpha-1,3-)-glycoprotein beta-1,4-N-acetylglucosaminyl-transferase isozyme B
	*MGAT5*	Mannosyl (alpha-1,6-)-glycoprotein beta-1,6-N-Acetyl-glucosaminyltransferase
Sialyltransferases	*ST3GAL3*	ST3 beta-galactosidealpha-2,3-sialyltransferase 3
	*ST3GAL4*	ST3 beta-galactosidealpha-2,3-sialyltransferase 4
	*ST3GAL6*	ST3 beta-galactosidealpha-2,3-sialyltransferase 6
	*ST6GAL1*	ST6 beta-galactosamidealpha-2,6-sialyltranferase 1
Neuraminidases	*NEU1*	Neuraminidase 1

**Table 3 ijms-20-04455-t003:** Demographics and disease activity parameters. Numbers before and in brackets indicate mean and range, respectively. SLEDAI-2K: SLE Disease Activity-Index-2000, anti-dsDNA: antibody to double-stranded DNA.

Subject Characteristics	Age	Female/Male	Disease Activity Parameter
SLE	42 (23–54)	17/1	
SLEDAI-2K			14 (6–30)
anti-dsDNA (IU/mL)			88 (2–220)
Control	54 (31–75)	17/2	

**Table 4 ijms-20-04455-t004:** Primer sequences used in the study for the mRNA amplification of glycosylation enzymes.

Name	Forward Primer	Reverse Primer
*RPL27*	5′-CGCAAAGCTGTCATCGTG-3′	5′-GTCACTTTGCGGGGGTAG-3′
*MAN1A1*	5′-TTGGGCATTGCTGAATATGA-3′	5′-CAGAATACTGCTGCCTCCAGA-3′
*MAN1A2*	5′-GGAGGCCTACTTGCAGCATA-3′	5′-GAGTTTCTCAGCCAATTGCAC-3′
*MAN2A1*	5′-CCTGGAAATGTCCAAAGCA-3′	5′-GCGGAAATCATCTCCTAGTGG-3′
*MAN2A2*	5′-TCCACCTGCTCAACCTACG-3′	5′-TGTAAGATGAGTGCGGTCTCC-3′
*MGAT1*	5′-CGGAGCAGGCCAAGTTC-3′	5′-CCTTGCCCGCAGTCCTA-3′
*MGAT4A*	5′-CATAGCGGCAACCAAGAAC-3′	5′-TGCTTATTTCCAAACCTTCACTC-3′
*MGAT4B*	5′-CACTCTGCACTCGCTCATCT-3′	5′-CACTGCCGAAGTGTACTGTGA-3′
*MGAT5*	5′-GCTCATCTGCGAGCCTTCT-3′	5′-TTGGCAGGTCACCTTGTACTT-3′
*ST3GAL3*	5′-TATGCTTCAGCCTTGATG-3′	5′-TTGGTGACTGACAAGATGG-3′
*ST3GAL4*	5′-ATGTTGGCTCTGGTCCTG-3′	5′-AGGAAGATGGGCTGATCC-3′
*ST3GAL6*	5′-TCTATTGGGTGGCACCTGTGGAAA-3	5′-TGATGAAACCTCAGCAGAGAGGCA-3′
*ST6GAL1*	5′-TGGGACCCATCTGTATACCACT-3′	5′-ATTGGGGTGCAGCTTACGAT-3′
*NEU1*	5′-CCTGGATATTGGCACTGAA-3′	5′-CATCGCTGAGGAGACAGAAG-3′

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
