# Peer review of "Altered Cell Surface N-Glycosylation of Resting and Activated T Cells in Systemic Lupus Erythematosus"

_ijms, 2019, doi:10.3390/ijms20184455_

Round 1
Reviewer 1 Report
The aim of this study was to examine T cell surface N-glycosylation in patients with systemic lupus (SLE). Peripheral blood mononulcear cells were isolated from 18 patients with SLE and 19 sex-matched healthy control subjects. Glycosylation status was assessed in resting and PHA-activated T cells. The results indicate that resting T cells from SLE patients exhibit activation-like phenotype as evidenced by greater binding of several plant lectins. Activated T cells isolated from SLE patients bind less galectin-1 than cells from control subjects but binding of other lectins is similar to control. Gene expression of several sialyltransferases and neuraminidase-1 are higher in activated T cells of SLE patients. Treatment of these cells with neuraminidase-1 restored galectin-1 binding. The results suggest that terminal sialylation of cell surface glycoproteins is higher in SLE T-cells making them less sensitive to galectin-1 induced apoptosis. This mechanism may contribute to immune system overactivity in this connective tissue disease.
The topic addressed in this study and the results are of interest. The manuscript is overall well-written. The only issue which needs further assessment is whether alterations of glycosylation are primary and contribute to the pathomechanism of SLE or secondarily result from the disease.
Reviewer 2 Report
Many thanks for writing an excellent paper - easy to understand, clear data - and with an interesting conclusion. It adds to several fields - SLE, inflammatory disease and T cell biology and was an easy paper to review. As far as I am concerned the paper should be accepted with minor revisions.
Line 4 - If authors contributed equally, why not in alphabetical order? No need to change - just to explain.
Line 43 - 'code' rather than code - for me, reads slightly better.
Line 48 - I know of papers which show Gal-1 binding to O-linked glycans - safer to either remove N, or write N and O.
Figure 1 legend. You have n numbers in your methods - can you place them here and under each figure? - makes it easier for the reader.
Line 101 - change '... various glycans depend...' to '...various glycans can depend...' In reference to papers that show Golgi pH, ER stress, substrate availability etc change glycosylation.
Lines 98-10 - You don't happen to have naive/CM/EM/TEMRA status for these T cells, do you? Perhaps, if it is known in the literature can you comment here or in the discussion about whether SLE patients have a higher number of TEMRA cells for example? You may be comparing cells with differing levels of antigen exposure. Not a major point, but I feel something needs mentioning.
Line 139 - Can the other ratios be shown as part of figure 3? I am a great believer in negative data - makes positive data more impactful.
Figure 4 - histograms need more controls. Can we need either no Gal-1 (ie background) or Gal-1 antibody alone, or binding + lactose (to block Gal-1 - any of these would do. You may need 2 histograms. Also, as mentioned before, n numbers would be great - I don't know if your error bars are technical or biological. Also, I assumed this was a healthy donor until I read the discussion - can you write that this is an SLE patient? In addition, if you have a comparison with a healthy volunteer this would help strengthen your conclusion.
Line 177 - best to say 'activation with PHA' as things may be different with other agonists.
Line 178 - This is where to bring in my earlier comment wrt the status of the cells - ie more likely to be TEMRA/EM in SLE compared to healthy controls (plus reference)
In the discussion there needs to be a reference somewhere to Siglecs - hypersialylation can lead to increased binding to Siglecs.
Lines 218-220 Need slight improvement - same style as rest of paper is needed.
Line 227 - do you have a university ethics code or number you can include?
Lines 232 and 235 - Thanks for emphasising no other inflammatory disease and DM - can you also say no other co-morbities?
Line 248 - '72h post-activation T-cell cultures...' would be better
Line 255 - was your Gal-1 for Vectorlabs as well? Can you state?
Lines 255-261 - confusion over methods - supplementary figure legends have viability dye first (which I believe is more common), whilst these methods have it last - can you reorder either?
Line 271 - did you treat cells with Neu in PBS? Also how many units of Neu did you use - conc?
SFig.3. Need untreated cells as well - readers need to see CD25 going up as evidence of activation.
